# A Comprehensive Review of Developments in Electric Vehicles Fast Charging Technology

Ahmed Zentani *, Ali Almaktoof and Mohamed T. Kahn

Department of Electrical, Electronic and Computer Engineering, Cape Peninsula University of Technology, Cape Town 7535, South Africa; almaktoofa@cput.ac.za (A.A.); khant@cput.ac.za (M.T.K.)
* Correspondence: 217001882@mycput.ac.za; Tel.: +27-747578467

**Abstract:** Electric vehicle (EV) fast charging systems are rapidly evolving to meet the demands of a growing electric mobility landscape. This paper provides a comprehensive overview of various fast charging techniques, advanced infrastructure, control strategies, and emerging challenges and future trends in EV fast charging. It discusses various fast charging techniques, including inductive charging, ultra-fast charging (UFC), DC fast charging (DCFC), Tesla Superchargers, bidirectional charging integration, and battery swapping, analysing their advantages and limitations. Advanced infrastructure for DC fast charging is explored, covering charging standards, connector types, communication protocols, power levels, and charging modes control strategies. Electric vehicle battery chargers are categorized into on-board and off-board systems, with detailed functionalities provided. The status of DC fast charging station DC-DC converters classification is presented, emphasizing their role in optimizing charging efficiency. Control strategies for EV systems are analysed, focusing on effective charging management while ensuring safety and performance. Challenges and future trends in EV fast charging are thoroughly explored, highlighting infrastructure limitations, standardization efforts, battery technology advancements, and energy optimization through smart grid solutions and bidirectional chargers. The paper advocates for global collaboration to establish universal standards and interoperability among charging systems to facilitate widespread EV adoption. Future research areas include faster charging, infrastructure improvements, standardization, and energy optimization. Encouragement is given for advancements in battery technology, wireless charging, battery swapping, and user experience enhancement to further advance the EV fast charging ecosystem. In summary, this paper offers valuable insights into the current state, challenges, and future directions of EV fast charging, providing a comprehensive examination of technological advancements and emerging trends in the field.

**Keywords:** electrical vehicles; DC fast charging; power electronics converters; control strategies

## 1. Introduction

The emergence of electric vehicles (EVs) has heralded a new era of environmentally friendly transportation, promising lower emissions and a cleaner environment. As EV use grows, the need for efficient and fast charging options becomes ever more critical. Fast chargers that are meant to provide speedy charging periods for electric car batteries have emerged as a possible answer to the problems encountered by EV drivers on longer excursions. These chargers provide a lifeline for travellers, allowing them to quickly recharge their automobiles while on the road, making long-distance electric mobility possible [1]. The requirement for fast battery charging has become a critical component in driving worldwide electric transportation growth. To encourage the widespread adoption of electric vehicles, charging choices that are dependable, convenient, and fast must compete with the refuelling experience of traditional internal combustion engines [2,3]. As a result, researchers and industry participants have made major efforts to develop cutting-edge charging technologies capable of dramatically reducing charging periods, hence increasing

the attraction and practicality of electric vehicles for consumers [4,5]. This study aims to shed light on the numerous fast charging technologies that have emerged in the electric vehicle ecosystem and power electronics converters and benefit drivers. The goal is to review the latest advancements in fast charging infrastructure, detailing widely used strategies, advantages, and disadvantages and also to discuss electronic power topologies and advanced control techniques aimed at reducing charging times and enhancing the efficiency of fast charging systems and explore the potential impact of fast charging on the EV industry. This study effectively establishes the importance of fast charging for EV adoption and justifies the need for this research in exploring and understanding current technologies and future trends. Electric vehicle (EV) fast charging falls into two categories: alternative current (AC) charging and direct current (DC) charging [6,7]. AC charging, which involves an on-board battery charger within the EV, tends to be slower due to limited power ratings [8,9]. Conversely, DC charging utilizes off-board battery chargers outside the vehicle, enabling faster charging speeds and greater power transfer capabilities [10,11]. As battery electric vehicles (BEVs) have become more widespread, the charging station design has rapidly evolved to accommodate these BEVs [12–16]. The speed at which EV battery packs charge is directly tied to the power transfer rate from the station, leading to the creation of fast and ultrafast charging stations capable of quick, high-power charging [17,18]. EV charging systems can allow power to flow in one direction or both directions [19]. Most on-board chargers facilitate one-way power flow, favouring grid-to-vehicle (G2V) charging [20]. This approach is preferred for its simplicity, reliability, affordability, and being easy to control [19]. Bidirectional chargers can feed power back into the grid through the grid-to-vehicle, V2G, mode [10]. Bidirectional charging is increasingly recognized for its potential to balance loads and integrate renewable energy, and reduce power losses in the grid [21–24]. This has sparked an increased interest in bidirectional chargers among researchers as a growing option for future EV applications. Charging units in AC bus-based architectures use separate rectifiers, enabling efficient and independent charging of multiple vehicles [25]. In contrast, systems with a common DC bus provide versatility and high-power operation [26,27]. Hybrid charging architectures, integrating AC and DC technologies alongside micro-grid systems, aim to optimize renewable energy use and enhance micro-grid performance beyond electric vehicle applications [28–30]. Researchers are dedicated to advancing charging stations through improved converters and smart control techniques to effectively manage the challenges of public charging. EV charging systems utilize multiple AC-DC and DC-DC converters and control strategies for safe and efficient battery charging, with the converter topology choice affecting the cost, size, performance, and efficiency of the system [31–34]. High-power converters can reduce charging time and provide additional grid services [25,35]. The increasing number of EVs and the incorporating renewable energy sources into the grid pose challenges to power quality, grid operation, safety, and reliability [36–38]. Researchers are developing various power converters, charging methods, and integration techniques to harness the benefits of EVs while addressing the challenges in this critical area for electric vehicle adoption. EV charging standards vary globally, with different regions adopting standards like SAE J1772, IEC 61851, and GB/T [26,38]. DC fast charging follows standards such as IEC-62196, CHAdeMO, CCS Type 1 and 2, and Tesla's proprietary standard. Standardization efforts are underway, including the ChaoJi standard [38]. While AC-connected systems maintain reliability, DC microgrids offer efficiency but face standardization challenges [39]. High-power DC fast charging may require grid upgrades, especially in rural areas and on highways [40,41]. Intelligent charging algorithms in modern chargers improve energy efficiency and grid stability, with energy storage systems aiding grid integration and cost reduction. Isolation from the AC grid is crucial in DC fast charging station design, and is achievable through low-frequency transformers or isolated DC-DC converters with careful consideration of soft-switching conditions [42].

The research emphasizes the importance of considering societal and economic factors when analysing fast charging infrastructure for electric vehicles (EVs). By exploring eco-

nomic feasibility, job creation, equity, and regulatory frameworks, the study provides a holistic understanding of the impact of fast charging systems, bridging technological advancements with broader societal objectives. Additionally, the paper offers valuable insights into current EV fast charging technology, covering techniques, infrastructure advancements, and challenges. Our research lays out pathways for the advancement of EV fast charging technology in the future, advocating for global collaboration, standardization, and advancements in battery technology to promote widespread EV adoption. This comprehensive approach underscores the significance of addressing societal, economic, and technological aspects to advance sustainable transportation through fast charging infrastructure.

The final section of the manuscript provides a comprehensive overview and analysis of EV fast charging infrastructure. It begins by examining various fast charging techniques and delving into advanced technical specifications, including charging standards and mode control. The paper also categorizes battery chargers for on-board and off-board charging, exploring converters and control strategies for EV systems. Challenges and future trends in EV fast charging and strategies to enhance efficiency and performance are discussed. The paper aims to contribute to the understanding and advancement of EV technology and fast charging infrastructure, offering insights to optimize solutions and improve energy utilization. It serves as a valuable resource for researchers and industry professionals, covering a wide range of topics from current state analysis to future trends and potential solutions.

## 2. Fast Charging Techniques for Electric Vehicles

Research and development efforts in fast charging techniques for electric vehicles are increasingly vital, aiming to reduce charging times and enhance the convenience of EV ownership [43]. According to the International Energy Agency's 2022 report, global electric car sales have surged, reaching 14% of all new car purchases, up from being 9% in 2021 and less than 5% in 2020. The first quarter of 2023 saw over 2.3 million electric cars sold, a 25% increase from the previous year. Projections suggest that 14 million electric vehicles will be sold by the end of 2023, with expectations that they may constitute 18% of total car sales for the year. Forecasts under the IEA Stated Policies Scenario indicate that electric car sales could reach 35% globally by 2030. This increasing popularity underscores the importance of efficient fast charging methods to support widespread adoption. Addressing the time needed to recharge EV batteries is a key challenge as EV adoption grows. A survey by McKinsey & Company found that 80% of EV owners consider fast charging availability to be crucial to their purchase decisions. Fast charging infrastructure is crucial for commercial sectors like taxi fleets and delivery services to minimize vehicle downtime and enhance operational efficiency. Significant advancements have been made in fast charging systems for electric vehicles (EVs) to meet the growing demand for high-power charging. These innovations enable shorter charging times compared to traditional methods. Key charging techniques include inductive charging, ultra-fast charging, DC fast charging, Tesla Superchargers, bidirectional charging V2G (vehicle-to-grid) integration, and battery swapping, as illustrated in Figure 1.

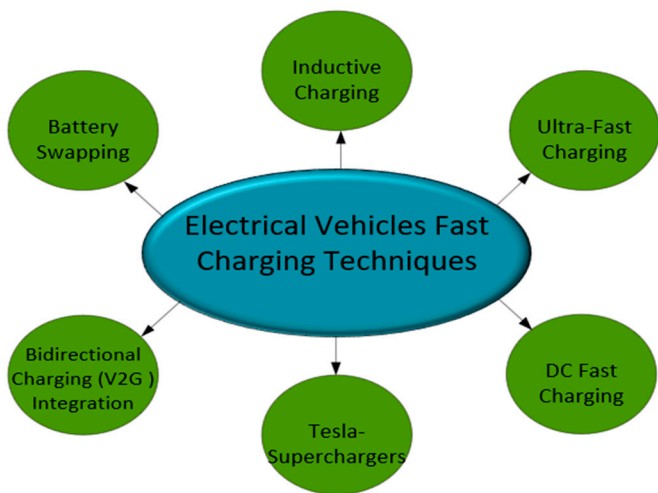

**Figure 1.** EV charging techniques.

### 2.1. Inductive Charging

Inductive charging, known as wireless charging, eliminates the need for physical cables by transmitting energy through an electromagnetic field between a charging pad on the ground and a receiver on the electric vehicle [44]. While not as fast as some wired fast charging methods, wireless charging offers convenience, allowing drivers to park over a charging pad for automatic charging without connecting any cables [44]. Nikola Tesla's experiment in 1910 at the Wardenclyffe Tower laid the groundwork for wireless power transfer (WPT) [45,46]; smart cities with e-mobility mainly utilize WPT [47], employing methods like inductive power transfer (IPT) and capacitive power transfer (CPT). Improving IPT efficiency involves optimizing coil design, ensuring coil alignment, selecting appropriate batteries and charging standards, and using electromagnetic field shielding [45,47–49]. Advanced optimization techniques and control strategies, such as adjusting charging frequency and monitoring alignment, enhance power transfer efficiency and safety. Inductive charging offers a convenient alternative to traditional plug-in charging methods, simplifying the connection process significantly. However, it is essential to consider potential safety risks, such as the possibility of foreign object hazards. While moderately weatherproof, inductive charging systems necessitate embedded charging pads for effective operation, influencing their integration with existing infrastructure. Moreover, compatibility is restricted to vehicles equipped with compatible receivers, posing a limitation on widespread adoption. Additionally, inductive charging may contribute to slightly accelerated battery degradation due to the heat generated during the charging process. Nonetheless, its scalability and upgradability are notable advantages, allowing for the incorporation of additional charging pads over time. It is worth noting, however, that inductive charging tends to exhibit a lower efficiency than traditional methods, which can be attributed to energy loss in the magnetic field.

Static charging systems, similar to IPT, are commonly found in stationary locations like parking lots, traffic lights, and toll booths, as depicted in Figure 2a. These systems are more efficient and compatible with electric vehicles. However, in wireless EV charging, misalignment between the transmitter and receiver coils can lead to fluctuations in the coupling coefficient, reducing the system's efficiency and power output [50]. Researchers are working to address misalignment issues in static WPT systems [50–53]. These systems typically have two stages: a high-frequency inverter converts input power to AC, which is then transferred to the primary coil to create an electromagnetic field. This induces AC power in the nearby receiver coil, which is then converted to DC power and stored in the vehicle's battery [45]. The system can achieve an efficiency of 85% or higher, with the individual stages reaching up to 97% efficiency.

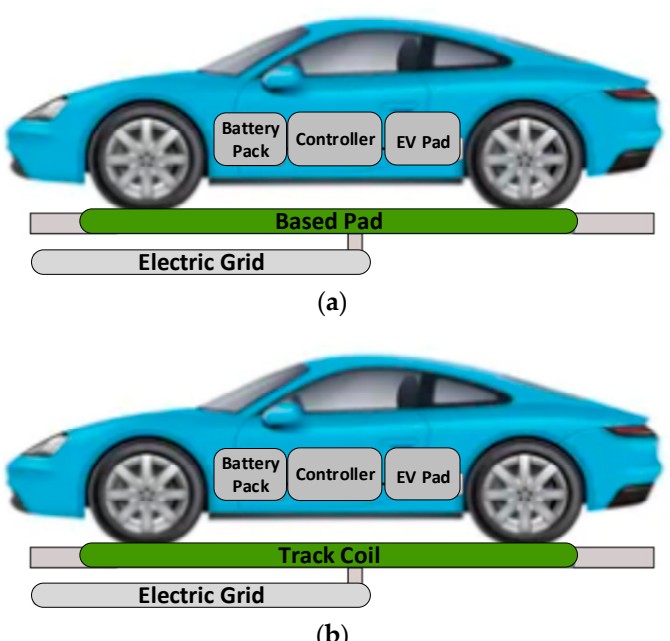

**Figure 2.** Inductive charging: (**a**) static charging. (**b**) dynamic charging.

Dynamic inductive charging is a method that wirelessly charges a vehicle's battery while it is in motion, also known as on-the-go or road-based charging. This addresses the challenge of electric vehicles having a limited range for long-distance trips [54,55]. It is an emerging technology that is currently being explored in various pilot projects and studies. In dynamic inductive charging, as depicted in Figure 2b, a primary coil buried along the road track and a secondary coil mounted on the vehicle chassis facilitate power transfer as the vehicle moves [47,56]. The primary infrastructure components include the Power Supply Unit and the Inductive Transmitter Units embedded in the road [47]. This charging method has several advantages; the major disadvantages are shown in Table 1.

### 2.2. Ultra-Fast Charging (UFC)

Ultra-fast charging (UFC) is a charging technology that significantly reduces EV charging times compared to traditional methods [57,58]. With power levels typically exceeding 350 kW, UFC delivers a lot of energy to the vehicle's battery quickly [57]. This approach addresses concerns about long charging times, making EV charging more comparable to refuelling a conventional vehicle and alleviating range anxiety [59,60]. UFC systems employ control strategies for efficient power delivery and battery management, including Constant Current (CC) and Constant Voltage (CV) modes and temperature monitoring. Although UFC stations are still in the early deployment stages and are less widespread than conventional infrastructure, their numbers are increasing, especially along major highways [61]. Both the charging infrastructure and EVs must be compatible with higher power levels, requiring station design and in-station and battery technology advancements. Automakers and charging providers are developing UFC technologies, introducing new standards like CCS and CHAdeMO 3.0 to support higher power levels [62,63]. Ongoing research aims to further improve charging speeds, infrastructure, and battery technologies in the dynamic field of UFC [61]. Ultra-fast charging offers unparalleled convenience with rapid charging times and minimal user interactions being required. However, it is important to acknowledge potential safety concerns, notably the risk of overheating. While generally safe, precautions should be taken to prevent overheating incidents. In terms of durability, ultra-fast charging stations are moderately weatherproof, ensuring reliable operation in various conditions. High-power charging stations facilitate integration with existing infrastructure, although their availability is not yet widespread. Compatibility primarily extends to newer electric vehicle (EV) models, though it is not guaranteed for

all makes and models. Additionally, the high charging currents associated with ultra-fast charging may contribute to slightly accelerated battery degradation over time. Nonetheless, the scalability and upgradability of these stations are notable advantages, as they can be modified to accommodate even faster charging speeds. Although ultra-fast charging systems boast high efficiency, some energy loss still occurs during charging.

### 2.3. DC Fast Charging (DCFC)

DC fast charging, also known as level 3 charging, is a prominent fast charging method that relies on high-power charging stations to deliver DC electricity directly to the vehicle battery, bypassing the onboard charger [18,64]. Unlike slower AC charging methods, which involve electricity passing through the vehicle's charger to convert AC power to DC, DCFC systems employ control strategies such as CC and CV modes and temperature monitoring for safe and efficient charging. These systems often incorporate intelligent power electronics and communication capabilities to optimize the charging process and grid integration. With the charging power typically ranging from 50 kW to 350 kW, DC fast charging drastically reduces charging times to as little as 30 min or less for a full charge, depending on factors like vehicle battery capacity and charging power level [65–67]. The global expansion of DC fast charging infrastructure, strategically placed along highways and urban areas [68,69], addresses range anxiety and promotes the feasibility of EV adoption. It is important to note that EV charging capabilities vary, with factors like vehicle model and battery technology influencing charging rates and maximum power support [70,71]. DC Fast Charging offers enhanced convenience with its faster charging times compared to standard AC charging methods, although it does require some user interaction. While generally safe, it is imperative to prioritize thermal management to mitigate potential risks. These charging systems are moderately weatherproof, ensuring a reliable performance under various conditions. Integration with existing infrastructure is widely available at dedicated charging stations but mandated DC fast charging equipment is required. The compatibility of this system spans most electric vehicle (EV) models, though it is not guaranteed to work for all makes and models. However, the faster charging speeds associated with DC Fast Charging may lead to slightly accelerated battery degradation compared to slower charging methods. Yet, the scalability and upgradability of these stations are notable, allowing for future upgrades to deliver even higher power outputs. Despite their higher efficiency, there is still some energy loss during the conversion process that is inherent to DC fast charging.

### 2.4. Tesla Superchargers

Tesla Superchargers are exclusively designed for Tesla vehicles and offer fast DC charging capabilities. These chargers are unique to Tesla and cannot be used with other electric vehicle brands [25]. Superchargers possess the ability to generate power outputs reaching a maximum of 250 kW, possibly adding up to 200 miles of range in just 15 min [72]. However, the charging rate experienced in practice can fluctuate due to various factors, including the battery's current charge level, battery temperature, and the capacity of the charging infrastructure [73]. Control strategies implemented in Tesla Superchargers include constant current and constant voltage modes, temperature monitoring, and battery management techniques. Tesla Superchargers also use advanced communication and vehicle-specific protocols to optimize the charging process and ensure compatibility with and safety for Tesla vehicles. The charging power levels of Superchargers have undergone advancements, resulting in newer versions that offer higher power outputs compared to their earlier counterparts. The charging time for Tesla vehicles utilizing Superchargers can also differ based on factors such as the specific model and battery size [74]. Superchargers can charge a Tesla vehicle to approximately 80% of its battery capacity in optimal conditions in about 20 min. However, it is essential to note that the charging rate may slow as the battery approaches total capacity to protect its health and longevity [75]. Tesla has been continuously expanding its Supercharger network, adding new charging stations and increasing the number of charging stalls at existing locations [76]. This expansion aims to provide

Tesla owners with convenient access to charging infrastructure, reduce range anxiety, and promote long-distance travel with electric vehicles. Tesla Superchargers epitomize convenience, offering an exceedingly user-friendly experience seamlessly integrated into the Tesla ecosystem. Safety is paramount, and there are rigorous safety protocols in place to ensure the well-being of users. Built to withstand diverse weather conditions, these chargers boast exceptional durability. However, access to the Tesla supercharger network is not universally available, necessitating reliance on Tesla's infrastructure. Compatibility is exclusively reserved for Tesla vehicles, thus limiting accessibility to owners of such vehicles. While the rapid charging provided by Tesla Superchargers may contribute to slightly faster battery degradation compared to slower charging methods, their extensive network, ongoing expansion, and potential for future upgrades underscore their scalability and upgradability. Despite their high efficiency, some energy loss occurs during the conversion process that is inherent to these chargers.

### 2.5. Bidirectional Charging Integration

Bidirectional charging, also known as vehicle-to-grid integration, is a promising concept in the electric vehicle ecosystem. It allows electric vehicles not only to draw energy from the grid for charging but also to discharge stored energy back into the grid [77], as shown in Figure 3. This bidirectional energy flow turns EVs into mobile energy storage resources, offering flexibility and support to the power grid [78]. Advanced bidirectional charging systems use communication protocols and control strategies to manage power flow, ensuring grid stability and optimizing energy usage. By participating in grid services, EVs with V2G capabilities can stabilize the grid, manage loads, and integrate renewable energy sources. This integration enhances grid stability, enables load management, and supports renewable energy integration, optimizing the use of renewable resources and improving grid efficiency [25,74,78,79]. However, successful V2G integration depends on a compatible infrastructure, communication systems, and regulatory support, which may vary by region and market dynamics [22,80].

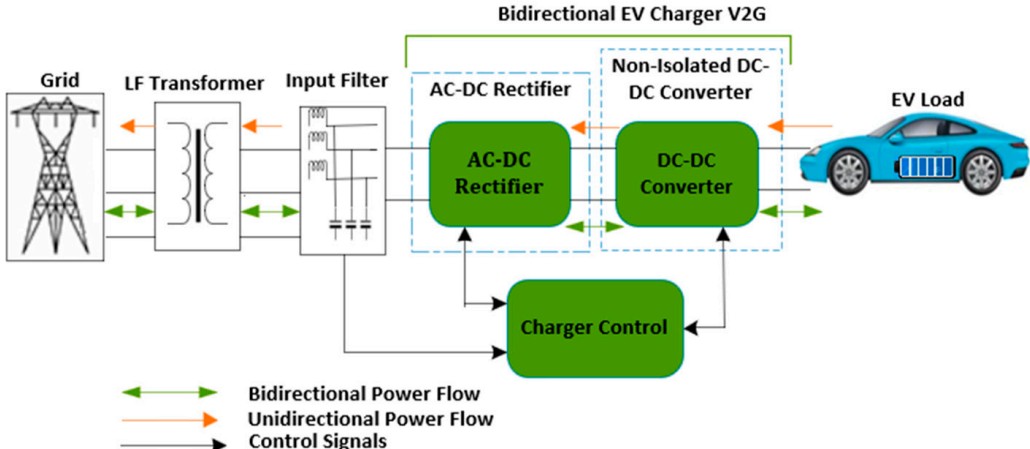

**Figure 3.** Bidirectional EV charging V2G.

Bidirectional charging offers convenience and demands additional setup and planning compared to standard charging methods. Although generally safe, its added complexity requires meticulous management. These chargers exhibit moderate weatherproofing, ensuring functionality in various conditions. However, integration with infrastructure relies on the availability of compatible charging stations and home setups, which are not yet widely accessible. Compatibility extends to some electric vehicles (EVs) with additional hardware, but it is not universally compatible across all models. The discharger cycles inherent to bidirectional charging may lead to slightly accelerated battery degradation over time. Nonetheless, there is potential for a broader adoption and future upgrades to enhance system capabilities, emphasizing its scalability and upgradability. Despite its advantages,

bidirectional charging tends to have lower efficiency compared to other faster charging methods, primarily due to additional conversion steps being involved in the process.

### 2.6. Battery Swapping

Battery swapping addresses the issue of long charging times associated with traditional methods by allowing drivers to exchange depleted batteries for fully charged ones quickly. Typically taking just a few minutes, this process benefits businesses managing fleets and individuals on long trips, reducing downtime [80], as shown in Figure 4. At dedicated swapping stations, the EV's discharged battery is replaced with a fully charged one, supported by advanced automation and communication technologies ensuring safe and efficient operations [81]. The swapped-out battery is then recharged for future use. Stations accommodate various EV models and employ automated systems for alignment and connections [82]. Advantages include reduced charging times, convenience, and potentially lower battery degradation [83]. While battery swapping offers a faster alternative, challenges like standardization and infrastructure costs must be addressed for widespread implementation, especially as charging technologies evolve. Battery swapping charging provides a faster alternative to standard AC charging, although it necessitates locating a swapping station. While generally safe, standardized procedures and automated mechanisms further ensure safety during swapping. These stations are designed for repeated swaps and offer moderate weatherproofing to withstand various conditions. However, their integration with existing infrastructure relies on the availability of a network of swapping stations with spare batteries, which is currently limited. Compatibility is contingent upon EVs being equipped with compatible battery packs, limiting universal applicability. Battery degradation may occur slightly faster due to handling and thermal stress during swaps. Despite the advantages, expanding a swapping network can be slower than upgrading charging stations, highlighting the differences in the scalability and upgradability of these techniques. Nonetheless, battery swapping boasts a high efficiency, although some energy loss occurs during handling and charging/discharging cycles.

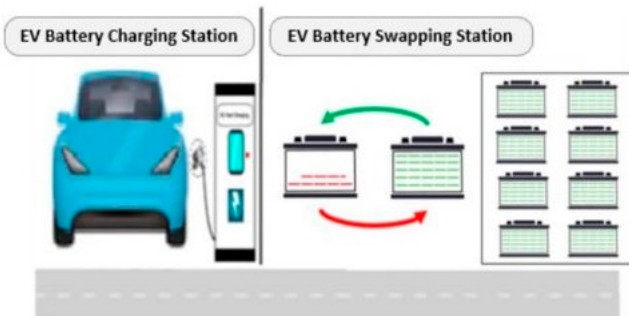

**Figure 4.** Battery swapping system.

Modular design in car manufacturing, especially with battery packs, presents a transformative opportunity for the industry. The concept of easily swappable battery packs alongside dedicated swapping stations could redefine the efficiency of networks and promote manufacturing adaptability. Recent advancements in assembly planning and Industry 4.0 technologies are making this modular approach feasible, potentially reshaping global production and factory flexibility in the automotive sector.

Table 1 provides a comprehensive overview of the specifications of these advanced fast-charging techniques for EVs, including their advantages and comparative analyses. The table covers several criteria essential to the effectiveness and viability of fast-charging systems, including convenience, safety, durability, integration with infrastructure, compatibility, battery degradation, scalability/upgradability, and efficiency. These criteria evaluate user-friendliness, safety measures, long-term reliability, compatibility with existing infrastructure, impact on battery health, ability to handle increased demand, and energy conversion efficiency.

**Table 1.** Electrical vehicles fast charging techniques vs. EV specs.

| EV Specifications | Ref | Inductive Charging Rating (1–10) | Ultra-Fast Charging Rating (1–10) | DC Fast Charging Rating (1–10) | Tesla Super-chargers Rating (1–10) | Bidirectional Charging Rating (1–10) | Battery Swapping Rating (1–10) |
|---|---|---|---|---|---|---|---|
| Convenience | [82–90] | (7) | (9) | (8) | (10) | (4) | (7) |
| Safety | [91–96] | (5) | (8) | (7) | (9) | (7) | (8) |
| Durability | [97,98] | (6) | (6) | (6) | (8) | (6) | (7) |
| Integration with Infrastructure | [25,61,96,99–105] | (4) | (5) | (8) | (7) | (5) | (4) |
| Compatibility | [89,92,102,106–110] | (3) | (7) | (7) | (3) | (5) | (7) |
| Battery Degradation | [61,107,109,111–117] | (5) | (6) | (5) | (5) | (5) | (5) |
| Scalability and Upgradability | [3,24,98,118–120], | (6) | (7) | (6) | (9) | (7) | (5) |
| Efficiency | [43,65,72,89,118,120–123] | (4) | (8) | (8) | (8) | (6) | (8) |

## 3. Advanced Infrastructure for DC Fast Charging for Electric Vehicles

A crucial aspect of the evolving electric vehicle landscape involves the development of advanced infrastructure for fast DC charging. This consists of implementing state-of-the-art systems and technologies that enable efficient and fast charging of electric vehicles [120].. Table 2 provides the power level charging ratings and classification standards of the different charging stations. The levels are categorized based on the supply system, maximum power rating (in kW), and maximum current rating (in A). DC fast charging, commonly referred to as Level 3 charging, offers much faster charging times [62]. It makes use of a higher-voltage power supply, often in the 400 to 800 volt range, which makes it possible for EVs to charge significantly more quickly, frequently providing up to 80% charge in as little as 20 to 30 min [62].

**Table 2.** Power level charging rating [121,122].

| Level of Charging Rating (A) | Supply System | Maximum Power Rating (kW) | Maximum Current Rating (A) |
|---|---|---|---|
| Level 1 (AC) (IEC) | 240 V | 4.7 | 16 |
| Level 2 (AC) (IEC) | 240 V | 11.5 | 32 |
| Level 3 (AC) (IEC) | 415 V | 90 | 250 |
| Fast DC Charging (DC) (IEC) | 600 V | 150 | 400 |
| Level 1 (AC) (SAE) | 120 V | 2 | 16 |
| Level 2 (AC) (SAE) | 208–240 V | 20 | 80 |
| Level 3 (AC)(SAE) | 300–920 V | 120-350 | 500 |
| Fast DC Charging (DC) (SAE) | −400 V | | |

The trend involves integrating renewable energy sources and energy storage systems into fast-charging networks to reduce their environmental impact and bolster sustainability. Planning approaches, simulation models, and optimization techniques are scrutinized to refine charging station locations and grid integration. Vehicle-to-grid (V2G) technology is gaining attention for its potential to ease the strain on the power grid and enhance reliability. However, detailed assessments of its operational mode in planned stations are limited. Integrating V2G into fast charging infrastructure planning is emerging as a strategy to optimize power system analyses, although current research often sidelines this aspect in favour of prioritizing losses and voltage stability [2].

Many important elements and factors contribute to the sophisticated infrastructure needed for DC fast charging. Below are several crucial aspects of the charging standards and charging modes control.

### 3.1. Charging Standards

The creation and observance of charging standards are essential elements in the field of advanced infrastructure for DC fast charging [62,109]. For owners of EVs, charging standards enable compatibility, interoperability, and seamless charging experiences across multiple charging networks and hardware providers [92]. The significant standards for EV charging systems shown in Table 3, along with some crucial points about the charging specifications for cutting-edge DC fast charging infrastructure.

**Table 3.** Major standards of EV charging systems [11,21,26,38,60,77,92,95,121,123–125].

| Standard | Description |
|---|---|
| IEC 60038 | Specifies the standard voltage levels used for electrical power systems and charging applications. |
| IEC 62196 | Standards conductive charging components for connectors, cables, outlets, plugs, inlets, and communication protocols used in AC charging of electric vehicles. |
| IEC 60664-1 | Insulation coordination for equipment within low-voltage systems. |
| IEC 62752 | Provides guidelines for connecting electric vehicles to information and communication technology, ICT, networks. |
| IEC 61851 | Covering various charging modes, communication protocols, and safety features. |
| SAE J1772 | Requirements for the electrical connectors and communication protocols for Level 1 and Level 2 charging used for AC charging of electric vehicles in North America. |
| SAE J2344 | Provides guidelines and test procedures for evaluating the crashworthiness and safety of electric vehicle battery systems. |
| SAE J2894/2 | Requirements for the power quality and conductive charge coupler used in DC fast charging electric vehicles. |
| SAE J2953 | Standards for interoperability to provide guidelines for conductive automated charging systems for electric vehicles. |
| SAE J2847/1 | Communication between vehicles as a distributed energy source and the grid. |
| SAE J3068 | Wireless power transfer for light-duty plug-in/electric vehicles and alignment methodology. |
| SAE J2931/7 | Evaluating the electrical performance of components used in hybrid and electric vehicles. |
| ISO 15118 | Standards for V2G communication protocols and interfaces between vehicle and charging infrastructure. |
| ISO 17409 | Specifications and reliable measurement of energy consumption allow for accurate comparisons and evaluations of different EV models. |

### 3.1.1. Organizations for Standardization

International bodies like the International Electrotechnical Commission (IEC), the Society of Automotive Engineers (SAE), and the International Organization for Standardization (ISO) are frequently responsible for developing and maintaining charging standards [97]. These groups work with industry participants to establish technical requirements and charging protocol standards for EVs.

### 3.1.2. Charging Connector Types

Various connector types and setups have been examined to ensure a dependable and safe link between the charging station and the electric vehicle. Commonly utilized connectors for DC fast charging include CHAdeMO and CCS [62] connectors [124]. These connectors guarantee compatibility between EVs made by various manufacturers and the charging infrastructure.

### 3.1.3. Communication Protocols

Communication protocols that enable data transmission between the charging station and the EV are also included in charging standards. These protocols allow for essential features to be controlled, including identification, control over power distribution, and real-time monitoring of charging conditions [125]. The Open Charge Point Protocol (OCPP) and ISO 15118 are two instances of communication protocols [126].

### 3.1.4. Power Level and Charging Speeds

The power levels and charging speeds enabled by the infrastructure are defined by charging standards. DC fast charging power levels can vary from 50 kW to several hundred kW, facilitating rapid charging durations [127]. Guidelines have established the upper limits for power and voltage to ensure safe and efficient charging procedures.

Table 4 provides a comprehensive overview of the specifications for commercial Plug-in Hybrid Electric Vehicles (PHEVs), Fuel Cell Electric Vehicles (FCEVs), and Extended-Range Electric Vehicles (E-REVs). Key details encompass a variety of vehicle models, classification by powertrain type, battery capacity measured in kilowatts (KW), driving range expressed in kilometres (KM), and connector types required for charging.

**Table 4.** Specifications of commercial electric vehicles [26,126,128–132].

| Category | Model | Type | Battery (KWh) | Range (Km) | Connector |
|---|---|---|---|---|---|
| Plug-in Hybrid (PHEV) | Chevrolet Volt | PHEV | 18.4 | 85 (battery) | Type 1 J1772 |
| | Mitsubishi Outlander | PHEV | 20 | 84 (battery) | CCS, Type 2 |
| | Volvo XC40 | PHEV | 10.7 | 43 (battery) | CCS, Type |
| | Toyota Prius Prime | PHEV | 8.8 | 40 (battery) | SAE J1772 |
| | Nissan Leaf | PHEV | 64 | 480 | CHAdeMO, Type 2 |
| Electric Vehicle (BEV) | Tesla Model S | BEV | 100 | 620 | Supercharger |
| | Tesla Model X | BEV | 100 | 500 | Supercharger |
| | Tesla Model 3 | BEV | 82 | 580 | Supercharger |
| Fuel Cell Electric Vehicle (FCEV) | Toyota Mirai | FCEV | 1.6 (hydrogen capacity) | 647 | N/A |
| | Hyundai Nexo | FCEV | 40 (hydrogen capacity) | 570 | N/A |
| | Honda Clarity | FCEV | 25.5 (hydrogen capacity) | 550 | N/A |
| Extended Range Electric Vehicle (E-REV) | BYD Atto3 | E-REV | 60.4 | 420 (battery) | CCS, Type 2 |

### *3.2. Charging Modes Control*

Charging modes are crucial in the field of modern fast charging infrastructure for maximising charging effectiveness and catering to the various needs of EV owners [126]. "Charging modes" refers to the many power distribution and charging methods used with DC fast charging infrastructure. Here, we present some essential information about charging modes.

### 3.2.1. Constant Current Charging

In this mode of control, the charging station feeds the EV battery a steady current while it is being charged. During the initial phases of charging, when the battery's state of charge (SoC) remains relatively low, this mode control method is frequently employed [127].

Constant current charging facilitates faster charge rates, ensuring prompt replenishment of the battery's capacity.

### 3.2.2. Constant Voltage Charging

As the battery's SoC reaches a predefined level, the charging station transitions to constant voltage charging. In this control mode, the charging station gradually reduces the charging current while sustaining a consistent voltage. Constant voltage charging is employed during the final stage to prevent overcharging the battery and ensure its longevity. Throughout this stage, the charging system continuously monitors the battery voltage and adjusts the current to prevent it from surpassing the designated level.

### 3.2.3. Constant Power Charging

The constant power charging mode adjusts the current and voltage to uphold a consistent power level throughout the charging procedure. This mode optimizes the charging speed and ensures an efficient utilization of available resources by dynamically adapting the charging conditions based on the battery's status and temperature [128]. When the battery's SoC is low, it enables quicker charging rates and automatically lowers the charging power when the battery gets close to capacity. Dynamic power control also enables load management and helps balance the power demand and supply within the charging infrastructure.

### 3.2.4. Demand–Response Charging

Demand–response charging capabilities may be included in advanced DC fast charging infrastructure. With this mode of control, the charging station can dynamically modify the charging power depending on the capacity of the grid and the electricity demand [129]. Reducing the charging power during heightened demand or when the electrical grid is strained allows for grid resources to be utilized efficiently.

### 3.2.5. Bidirectional Flow Charging

Bidirectional power flow is supported by cutting-edge fast charging infrastructure, allowing energy to be transmitted to and from the EV's battery. By employing this capability, EVs receive grid-based charging and discharge energy back into the grid or supply power to other consuming devices [130]. Vehicle-to-grid applications, where EVs can help to stabilise the grid and offer grid services, may be supported via bidirectional charging [131]. Different charging modes are implemented in sophisticated DC fast charging infrastructures, giving EV owners and grid operators flexibility and optimisation options based on battery state, grid limitations, and user choices; it also enables efficient charging [92]. EVs may be charged quickly while preserving battery health, grid stability, and economical power usage by utilising the correct charging mode. Furthermore, intelligent charging algorithms and communication protocols that enable real-time monitoring, control, and coordination of the charging process are frequently used in conjunction with charging modes [132]. These characteristics facilitate grid integration, optimise power distribution, and guarantee secure and dependable charging operations. Implementing charging modes within advanced DC fast charging infrastructure enhances the user experience for electric vehicle owners during charging sessions. Additionally, it promotes optimal energy utilization and contributes to the broader adoption of EVs as a sustainable transportation option [133].

### 3.2.6. Temperature Monitoring and Control

DC fast charging produces heat because of the substantial charging currents it employs. Temperature monitoring and control techniques have been implemented to safeguard the battery against overheating. Temperature sensors are positioned inside the battery pack to monitor its temperature throughout the charging process [134]. The charging system utilizes these data to modify charging parameters like current or voltage, ensuring the battery remains within safe temperature thresholds. This practice is crucial for preserving optimal

battery performance and extending its longevity [135]. The good news is that advancements in lithium-ion batteries continue to be made, opening doors for next-generation options like solid-state batteries. Material research is key to overcoming challenges and making electric vehicles more sustainable. By focusing on improving the heat resistance of materials, using venting mechanisms, and employing eco-friendly bio-based flame retardants, researchers are working to prevent thermal runaway, a critical safety concern.

### 3.2.7. State of Charge (SoC) Estimation

Accurate estimation of the battery's SoC is crucial for efficient charging. Control strategies utilize algorithms and models to estimate the SoC based on various parameters such as voltage, current, and temperature measurements [136–138]. These methods consider the battery's discharge and charge characteristics to provide a reliable estimate of its state of charge [136]. SoC estimation helps regulate the charging process, ensuring the battery is neither undercharged nor overcharged [139].

## 4. Electric Vehicle Battery Chargers Categories

Electric vehicle battery chargers can be classified according to their charging capabilities, physical configurations, and intended applications as on-board or off-board chargers. On-board chargers are incorporated within the vehicle, and their capacity determines their charging rate. They are commonly found in electric vehicles and are compatible with various charging infrastructures. On the other hand, off-board chargers are external units, and their specifications determine the charging rate. They are used in public charging stations and private charging points where on-board charging is not possible or convenient. On-board charging, while convenient for many EV owners, may not always suffice in certain situations, leading to the need for off-board chargers. Challenges such as long-distance travel, lack of personal charging facilities, and the demand for faster charging options highlight the limitations of on-board charging. Off-board chargers, including fast-charging stations along highways, public locations, and commercial premises, address these challenges by offering quick and accessible charging solutions for EV users.

### 4.1. On-Board Charging

On-board charging (OBC) in electric vehicles involves converting external AC power into DC power to charge the vehicle's battery pack. It is a critical process requiring various components like rectifiers and transformers to ensure safe and efficient charging [62]. The OBC, typically integrated into the vehicle's powertrain or near the battery pack, can be built within the vehicle or be part of the charging cable [140], as shown in Figure 5. Common methods include Level 1, standard outlets, and Level 2, with dedicated charging stations offering higher voltage. On-board charging provides convenience, as EVs can be charged at home, work, or other accessible locations. Level 1 charging (120 volts) is cost-effective, while Level 2 (240 volts) provides flexibility without requiring specialized infrastructure [3]. Owners can charge their vehicles wherever suitable outlets or stations are available [141]. For on-board chargers, popular models and manufacturers commonly found in electric vehicles include the Tesla Supercharger, which offers fast-charging capabilities for Tesla owners, the Nissan CHAdeMO which is compatible with the Nissan Leaf for fast charging at CHAdeMO-equipped public charging stations, the BMW i3 equipped with an on-board charger supporting both AC and DC charging for flexibility at various public charging stations, and the Chevrolet Bolt EV with an on-board charger capable of accepting both AC and DC charging for fast-charging at compatible public charging stations. Table 5 details the advantages and disadvantages of on-board and off-board chargers.

**Table 5.** Comparison between on-board and off-board chargers [142].

| Charging Methods | Advantage | Disadvantage |
|---|---|---|
| On-board charger | - A lower rate of energy transfer (KW).<br>- Convenience (charge anywhere with an outlet).<br>- Weight is added to the EV.<br>- An on-board rectifier controls battery management system (BMS). | - Slow charging times.<br>- Added weight and size to the vehicle.<br>- Limited charging infrastructure compatibility.<br>- May require specific outlets.<br>- Potential battery heating issues. |
| Off-board charger | - High energy transfer (KW).<br>- The problem of battery heating must be solved.<br>- Potential enhanced BMS systems.<br>- Fast charging speed.<br>- Charger at higher power levels.<br>- Reduces weight in the vehicle. | - Limited flexibility (requires compatible charging stations.<br>- Potential challenges with the battery management system.<br>- Higher cost and complexity.<br>- Charging station accountability.<br>- Limited ability to identify defective battery cells. |

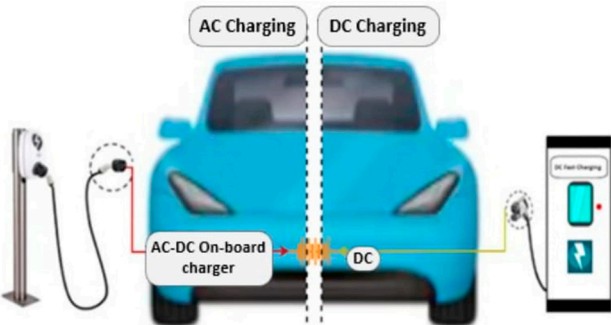

**Figure 5.** Illustrated on-board/off-board EVs fast charging.

*4.2. OFF-Board Charging*

Off-board charging involves replenishing EV batteries outside the vehicle itself, typically at external charging stations or infrastructure [143], as shown in Figure 5. Researchers are exploring off-board and DC-based charging methods for EVs due to their potential impact on power quality and efficiency [137]. Common off-board charging methods include public charging stations, DC fast charging (Level 3), and wireless charging (inductive charging) [117]. These methods have fewer conversion stages, making charging more efficient, especially with a DC power supply, which eliminates the need for AC-DC conversion and power factor correction stages. While DC chargers are not widely adopted in EVs yet, they offer the potential for the integration of solar PV power. Some studies propose dual-input EV chargers that can utilize both AC grid power and standalone PV systems, operating in grid-connected mode during low solar irradiation and in V2G mode during idle hours [138]. Off-board chargers can be divided into isolated and non-isolated converters, with isolated converters being preferred at higher voltage levels and non-isolated topologies being feasible for low power levels [65,144,145]. Off-board chargers include examples such as the Tesla supercharger network, strategically located along highways and major routes to offer high-speed charging for Tesla vehicles during long-distance travel, charging points providing both Level 2 AC chargers and DC fast chargers, which are commonly found in public parking lots and commercial premises for convenient charging options, various industries currently provide fast-charging stations supporting multiple charging standards like CCS and CHAdeMO, which are primarily located along major highways and urban areas, and home charging stations, installed by many EV owners at their homes for convenient overnight charging or charging during low electricity demand periods, typically including Level 2 AC chargers and being installed in residential garages or driveways.

## 5. Status of DC Fast Charging Stations and DC-DC Converters Classification

Efficiently and quickly charging electric vehicles demands high-power DC-DC converters to adjust the charging infrastructure's high-voltage DC power to the battery's required voltage. Various converter topologies are used, with recent studies proposing designs with fewer active and passive components [146]. To minimize switching losses, soft-switching power electronic switches are being introduced [147,148]. These converters, used in both on-board and off-board chargers, include buck, boost, buck–boost, SEPIC, Cuk, Zeta, and Super-lift Luo converters [24–26].

Higher power chargers typically employ isolated DC-DC converters with options like fly-back, forward, push–pull, half-bridge, full-bridge, and multilevel converters [144,149]. The bidirectional operation of the transformer is achievable in multiple switch topologies through the alternate operation of the switches [65]. However, this approach has limitations, including transformer core saturation and the strain on primary-side switches caused by operating in discontinuous mode [145]. Newer topologies, like bridgeless and interleaved configurations, aim to enhance efficiency, reduce voltage/current ripples, and streamline the number of components. Soft-switching techniques with resonant converter topologies may further boost charging efficiency [150]. For high-gain applications, multilevel converters are another multi-switch alternative. Additionally, their use guarantees less electromagnetic interference (EMI) because there are fewer voltage jumps [146]. Prioritizing EV charger design effectiveness and charging time is crucial, considering most EVs require 3–4 h to charge fully, a critical factor for their potential use in public transportation [147]. Figure 6 outlines a general classification of DC-DC converter topologies for EV powertrains.

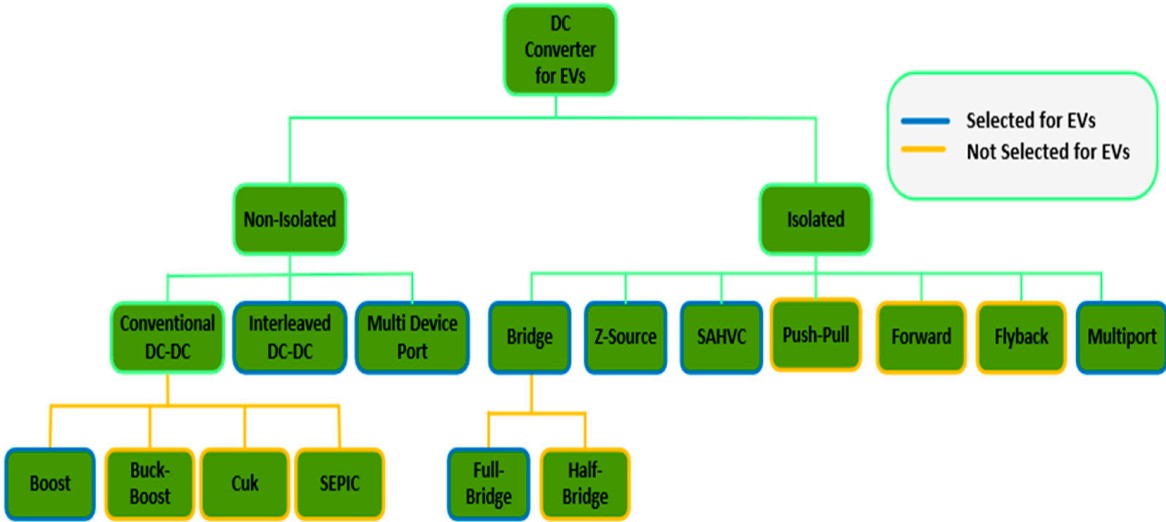

**Figure 6.** Classification of DC-DC converters utilized for EV charging systems.

Figure 7 illustrates the key components of a typical DC fast charger power system, featuring two conversion stages for converting three-phase AC power to DC, along with a DC-DC stage incorporating galvanic isolation [121]. The AC-DC rectification stage includes a Power Factor Correction circuit to meet grid codes' power quality requirements. The DC-DC stage facilitates parallel connectivity at the charger's output stage, ensuring isolation between the EV and the grid [121]. Two main approaches for achieving galvanic isolation are presented: One method involves utilizing a low-frequency transformer (LFT) positioned between the grid and the AC-DC stage, as depicted in Figure 7a. These techniques have been extensively discussed in various sources [66,149,150]. Alternatively, a high-frequency transformer (HFT) can be incorporated within the DC-DC stage, as shown in Figure 7b. These methods have also been explored in multiple references. Figure 7 visually represents the two isolation options, focusing on a single module charger for simplicity. However, as the power requirements of DC fast chargers increase, the system's output power can

be enhanced by connecting multiple identical modules in parallel. This approach enables the system to meet the growing demand effectively. For those seeking more detailed information and comprehensive insights into LF and HF transformer configurations, we advise you to refer to the aforementioned references [66,144,148–151]. The article proposes integrating a filter at the input stage to address the harmonic distortion originating from the rectifier. This filter aims to mitigate the inherent harmonic distortion in the current drawn by the rectifier. LC or LCL filters are typically favoured over L filters due to their superior performance characteristics. Additional information on the advantages and specifications of LC or LCL filters can be found in references [152–155].

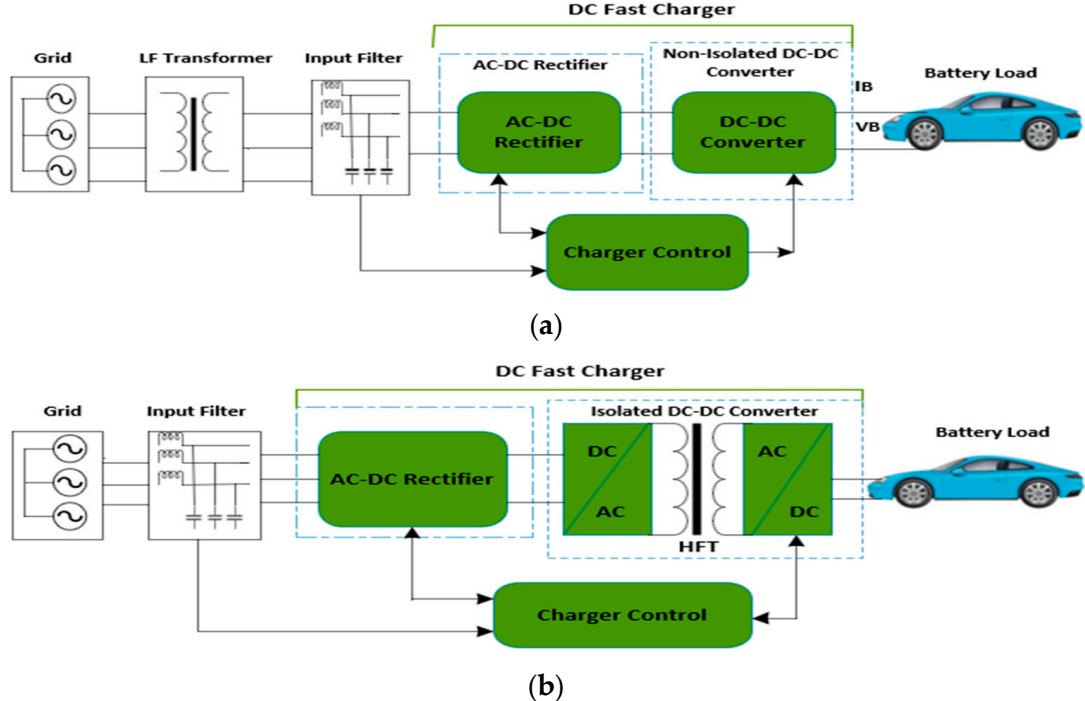

**Figure 7.** Block diagram of DC Fast Charging Power Conversion Stage with: (**a**) LFTS Configuration. (**b**) SST-Based HFT Configuration.

## 6. Control Strategies for EV Systems

This section focuses on the control technology employed in DC-DC converters for EV fast charging. It covers control objectives, methods, and challenges associated with different converters. Various control techniques such as voltage mode, current mode, peak current, and average current control are examined, applicable to isolated and non-isolated converter systems. Table 6 offers a comparison of control strategies for these converter types.

Control objectives in DC-DC converter control technology refer to the desired performance criteria that need to be achieved. These objectives typically include regulating the output voltage or current, achieving a fast response to load changes, maintaining stability, minimising ripple and noise, and maximising efficiency. Ongoing research and development efforts continue to refine these control technologies and explore new approaches to optimize charging performance, integrate them with smart grids, and enhance user experience. As the EV market expands, advancements in control technologies will play a vital role in shaping the future of fast and reliable charging infrastructure. Control methods in DC-DC converter control technology encompass the techniques and algorithms to achieve the desired control objectives. Some commonly employed methods include voltage mode control, current mode control, peak current control, and average current control. Each method has advantages and trade-offs regarding response speed, stability, and complexity. However, despite the availability of various control methods, DC-DC converters can encounter common problems that need to be addressed. These issues include

output voltage or current overshoot or undershoot, instability leading to oscillations or ringing, poor transient response, excessive ripple or noise, and cross-regulation problems in multiple-output converters. Effective control strategies should consider these challenges and aim to mitigate them.

In voltage mode control for non-isolated and isolated DC/DC converters, charging techniques include pulse width modulation, peak and average current mode control, feedforward control, proportional–integral–derivative control, and predictive control. Techniques for non-isolated DC/DC converters under current mode control involve current sensing, cycle-by-cycle current limiting, and slope compensation. The peak current control technique in non-isolated converters employs peak current sensing and limiting, while average current control utilizes the average current sensing and limiting control techniques. Isolated DC/DC converters under voltage mode control use the same methods as non-isolated converters. In contrast, the current mode control technique utilizes current sensing, cycle-by-cycle current limiting, and slope compensation. Peak current control in isolated converters employs peak current sensing and limiting, while average current control uses average current sensing and limiting. These strategies and techniques are tailored to ensure effective voltage and current regulation in various DC/DC converter configurations.

When choosing a control method, considering the impact of charging methods on battery durability, it is crucial to prioritize factors like battery health, longevity, and service life. To achieve this, it is necessary to assess how each method affects battery degradation and lifespan [156]. This evaluation should consider aspects such as charging rate (high currents degrade batteries faster), temperature management (cooler temperatures promote battery health), and general stress on the battery cells during charging [157]. Prioritizing these factors in the charging process can extend the lifespan of electric vehicle batteries and maximize their long-term performance. For optimal charging profiles, always consult the battery manufacturer's specifications. Remember that cooler temperatures and multi-stage charging can enhance battery life. Ultimately, the best control method for battery life requires balancing efficiency, speed, and the battery's characteristics.

**Table 6.** Control strategies for EV systems techniques for isolated and non-isolated DC-DC converters [10,158–161].

| Control Methods (Non-Isolated) | Benefits |
| --- | --- |
| Voltage Mode Control | Improves charging efficiency, fast response, regulation protects the battery. |
| Current Mode Control | Enhancing charging efficiency, ensuring the charging current remains within a predetermined threshold, reducing charging times, minimizes energy losses, and improves efficiency. |
| Peak Current Control | It prevents overloading the converter and charging infrastructure, regulates and limits the peak current to optimize the charging process, prevents the current from exceeding the peak limit, ensures efficient energy transfer, and provides overcurrent protection. |
| Average Current Control | Optimizes energy transfer and improves charging, ensures optimal charging performance and power quality, and reliable, safe, fast charging operations. |
| **Control Methods (Isolated)** | |
| Voltage Mode Control | Maintaining a stable output voltage provides efficient energy transfer, stability from the grid to the EV battery, and compatibility with charging infrastructure. |
| Current Mode Control | Provide efficient energy transfer, stability from the grid to the EV battery, fast dynamic response to load variations, and excellent adaptability to load variation conditions. |
| Peak Current Control | Regulates and maintains a stable average output current, demand for rapid and reliable EV charging increases, compatibility with grid constraints, maintaining safety, optimizing energy transfer, and increasing charging efficiency demand for EV fast charging. |
| Average Current Control | Regulates and maintains a stable average output current; it ensures compatibility with charging infrastructure and provides flexibility in adapting to changing load conditions, allowing the system to accommodate different charging scenarios, optimizing energy transfer, reducing charging times, and creating reliable and high-performance charging infrastructure. |

### 7. Challenges and Future Trends in EV Fast Charging

EV fast charging systems are continuously evolving to meet the demands of the growing electric mobility landscape. Challenges and future trends in EV fast charging revolve around addressing infrastructure limitations, including expanding the charging network and managing grid capacity intelligently through technologies like load balancing and demand response. Universal standards and interoperability among charging systems are crucial for widespread EV adoption, requiring global collaboration to establish compatibility across networks and countries. Despite advancements, battery technology poses challenges, necessitating improvements in chemical compositions and thermal management for faster charging without sacrificing battery longevity. Future research areas recommend prioritizing faster charging, improving infrastructure components, standardizing charging processes, and optimizing energy utilization through smart grid solutions and bidirectional chargers. Exploration of battery advancements, wireless charging, battery swapping, and enhancing user experience is encouraged. Enriching the current trends involves detailing advancements in ultra-fast charging for quicker charging times, exploring battery technology advancements, emphasizing standardization efforts for compatibility across different EVs and charging stations, and highlighting the integration of renewable energy sources with charging stations for a more sustainable ecosystem.

This research paper on EV fast charging technology highlights key areas for recommended future research, emphasising the need to prioritise faster charging, improve infrastructure components, standardize charging processes, and optimise energy utilization through smart grid solutions and bidirectional chargers. It encourages exploration of battery advancements, wireless charging, battery swapping, and enhancing user experience. Future research should comprehensively analyse the impact of EV fast charging on the power grid, delve into DC fast charging, ultra-fast charging, and V2G systems, advance infrastructure, and control techniques, integrate renewable energy sources, address standardisation challenges, implement intelligent charging algorithms, and strategies for efficiency enhancement. Achieving faster charging times and exploring various charging techniques, control strategies, and using EV batteries for grid services are crucial areas for ongoing exploration.

### 8. Research Contribution

This research paper makes insightful contributions to the field of EV fast charging, standing out for its comprehensive analysis of crucial aspects. It offers a multifaceted contribution that propels us towards a more electrifying future. The paper's impact on the field can be summarised in three key ways:

- Comprehensive Landscape Analysis: It paints a complete picture, delving into diverse charging categories, methods, infrastructure specifics, and crucial elements like charging modes control, standards, converters, and control strategies. This holistic view informs future research and development efforts.
- Detailed Comparative Evaluation: By meticulously analysing various charging techniques, including their advantages and limitations, the paper empowers informed decision-making for future infrastructure advancements and technology choices.
- Challenges and Future Outlook: The article does not shy away from addressing current limitations and boldly proposes future research directions. This forward-thinking approach paves the way for overcoming obstacles and achieving faster, more efficient, and sustainable EV charging solutions.

### 9. Conclusions

This paper has thoroughly examined the latest advancements in fast charging infrastructure, elucidating widely adopted strategies alongside their advantages and disadvantages. It has delved into power electronic topologies and advanced control techniques to reduce charging durations and enhance efficiency within fast charging systems. Additionally, exploring the potential impact of fast charging on the electric vehicle (EV) industry

has underscored its significance for EV adoption. This research highlights the critical need to understand current technologies and anticipate future trends to effectively navigate the evolving landscape of fast charging and its implications for the broader EV ecosystem. It aims to bring clarity to the pursuit of faster charging times, emphasising high-power fast charging as a significant focal point of EV charging technology. The review explores methods to enhance the efficiency and effectiveness of fast charging systems, providing insights into current developments in EV fast charging techniques and power electronic converters utilized in fast-charging systems. Control strategies for electric vehicles are analysed, alongside the impacts of high-power fast charging on battery systems and future developmental trends. Various fast charging methods for electric vehicles are extensively examined, including inductive charging, UFC, DCFC, Tesla superchargers, V2G integration, and battery swapping, with a summary of the associated advantages and drawbacks for each technique provided. Moreover, the essential components and considerations involved in the intricate infrastructure for DC fast charging are outlined. An analysis of high-power rear DC-DC converters, both isolated and non-isolated, utilized in on-board and off-board chargers has been conducted. These two-stage or single-stage converters efficiently convert high-voltage DC power for the fast-charging infrastructure of electric vehicle batteries. Control strategies employed for these converters are also investigated, with the primary objective being to maintain consistent performance across the entire load spectrum, minimize current fluctuations, and enhance system efficiency. The efficiency, affordability, safety, reliability, and control mechanisms of converter configurations have significantly influenced the advancement of fast charging technology for electric vehicles. Fast charging systems integrated with V2G capabilities can lower charging expenses and enable EVs to engage in grid services, extending their interaction with the electrical grid beyond conventional charging. When connected to the grid, EV batteries can serve various purposes, including grid stabilization, load management, and integrating renewable energy sources.

**Funding:** This research received no external funding.

**Conflicts of Interest:** The authors declare no conflict of interest.

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
