# Peer review of "A Comprehensive Review of Developments in Electric Vehicles Fast Charging Technology"

_applsci, doi:10.3390/app14114728_

Round 1

Reviewer 1 Report

Comments and Suggestions for Authors

The Figs in the manuscript are not clear, authors should provide clear figs in the manuscript.

Comments on the Quality of English Language

The English editing is fine for publication.

Author Response

Dear Reviewer,

Thank you very much for taking the time to review this manuscript. Please find below a detailed summary of the revisions made to the manuscript titled "A Comprehensive Review of Electric Vehicles Fast Charging Developments and Technology" in response to the referees' comments. We have carefully addressed each point raised and made the necessary improvements to enhance the quality of the paper.

Reviewer 2 Report

Comments and Suggestions for Authors

This review paper thoroughly investigates the development of fast charging technology for electric vehicles (EVs), including its advantages and comparative analyses from various perspectives. EV charging techniques were listed in figure 1.  Here are some specific comments I would like the authors to address.

1.     Each technique explained briefly in this article, some cited references might be descripted clearly for these techniques. Please list them for easy to study these techniques.

2.     In Table 1, the words such as medium to low and medium to high are difficult to understand the degree, an exacted number may be easy to know. For example, number 1 to 10.

3.     In Table 6, the control strategies were presented but these were explained too rough to know which kinds of charging techniques used these control strategies.

4.     Please correct the quotation marks in line 42 and double check and modify these tiny errors.

Many citied references were published in 5 years. Authors used tables to compare each method is a good way to present just need some modifications. Finally, the conclusion is easy to understand. Overall, I would recommend a major revision.

Author Response

(The authors gave the same response as above.)

Reviewer 3 Report

Comments and Suggestions for Authors

The manuscript entitled "A Comprehensive Review of Electric Vehicles Fast Charging Developments and Technology" is a review work. It reports the interesting and important topic of EV fast charging technologies.

The structure of the EV fast charging system and the method of its control are listed an characterized. The selection of presented issues and the way they are reported depend on the authors' point of view. It may or may not be considered correct and up-to-date. In opinion of this reviewer, the review presented is written clearly and completely.

Two comments, relevant to this reviewer, are presented below.

Page 15, Table 6. Comment: It is very important to optimize the use of batteries from the point of view of their durability. Therefore, it is worth considering the impact of the method of charging batteries on their service life. Perhaps it would be useful to make some reference in Table 6 to Table 1, which highlights this problem.

Page 16, Section 7. Challenges and Future Trends in EV Fast Charging. Comment: Thanks to extensive work related to the literature review, the authors gained extensive knowledge about development trends in the field of fast charging technologies and systems relevant for electric vehicles. It is worth highlighting this acquired knowledge here. Therefore, in this section it would be appropriate to present current trends in more detail and cite (again) relevant literature.

Conclusion.

In the opinion of this reviewer, the manuscript “A Comprehensive Review of Electric Vehicles Fast Charging Developments and Technology " meets the requirements for publication in the "Applied Science" journal. It may be published after the minor revision.

Author Response

(The authors gave the same response as above.)

Reviewer 4 Report

Comments and Suggestions for Authors

The submitted manuscript deals with the reviewing of the Electric Vehicles Fast Charging Developments and Technology. The authors have done an extensive literature review on the subject. However, the paper needs more modifications and additions from my point of view.

1-     The abstract can be revised with better consideration of the main findings. The majority of the abstract is introductory, which should be shortened. Moreover, the main concluding remarks regarding the EVFC, its advantages, gaps... as well as future prospects can be added.

2-     The introduction part is well elaborated. However, it does not accurately describe the objectives and justification of this work. Even in a review article, some issues should be described in the text. Issues: What is the need to do a review article on this topic? What is the great contribution of this article to researchers or industry?

3-     Introduction part: In the final paragraph, explain the structure of the manuscript, i.e., explain how the paper is organized.

4-     Numerous relevant papers have been published in recent years, especially in  2023 and 2024. Some key, important or/and latest research results in this topic should be discussed and cited in corresponding sections.

5-     The information presented in part 2 and part 3 in this paper are not very new and limited in scope. They appear to be a simple compilation of results with no critical analysis and very few comments. Thus, these parts in this paper needs deep discussion with presenting the authors' viewpoints in these parts.

 6-     Some sections are severely lacking in content.

 7-     In section 4: It would be helpful to provide specific examples or commonly used chargers for each category to enhance the reader's understanding. For on-board chargers, mention some popular models or manufacturers commonly found in electric vehicles. Similarly, for off-board chargers, mention examples of public charging stations or private charging points where these chargers are typically used.

 8-     Elaborate on the reasons why on-board charging may not always be possible or convenient, leading to the need for off-board chargers. Highlight scenarios such as long-distance travel, lack of access to personal charging facilities, or the need for faster charging options that necessitate the use of off-board chargers.

 9-     In the Challenges and Future Trends in EV Fast Charging, briefly mention potential areas for future research and development in the field of EV fast charging. This could include exploring emerging technologies, addressing specific technical challenges, or considering the integration of renewable energy sources into fast charging infrastructure.

 10- The review should place prior original research of the authors (and/or their group/environment) in context, clarifying what the authors have contributed with to the research topic, without exaggerating.

11- Review papers should present novel insights or conclusions for directing the respective research area(s). Please add appropriate statements in the abstract, Introduction section, and Conclusion section.

12- The quality of the figures and charts is very bad. Please make effort to bring the presentation and quality of your figures, photos, and tables to the very best. Please make an effort to bring the presentation and quality of your figures, photos, and tables to the very best. 

13- The review would benefit from being expanded. It is more useful as a literature survey with more extended referencing. In a review, it is essential to provide an intellectual contribution/insight/outlook beyond what is already covered in existing reviews (if any; they should be cited) on the same or closely related topics. This is probably here, but it should be explicitly clear in the paper (usually but not always in the introduction) how this is achieved.

Comments on the Quality of English Language

Moderate editing of English language required

Author Response

(The authors gave the same response as above.)

Round 2

Reviewer 4 Report

Comments and Suggestions for Authors

Change the final paragraph of the introduction section to
#Overall, this paper provides a comprehensive overview and analysis of EV fast charging infrastructure. It begins by examining various fast charging techniques and delving into advanced technical specifications, including charging standards and mode control. The paper also categorizes battery chargers for on-board and off-board charging, exploring converters and control strategies for EV systems. Challenges and future trends in EV fast charging and strategies to enhance efficiency and performance are  discussed. Thus, the paper aims to contribute to the understanding and advancement of EV  technology and fast charging infrastructure, offering insights to optimize solutions and improve energy utilization. #

Comments on the Quality of English Language

Minor editing of English language required